# Distance Learning in Nursing Education during the COVID-19 Pandemic: Psychosocial Impact for the Greek Nursing Students—A Qualitative Approach

**DOI:** 10.3390/healthcare11081178

**Published:** 2023-04-19

**Authors:** Evangelia Kartsoni, Nikolaos Bakalis, George Markakis, Michail Zografakis-Sfakianakis, Evridiki Patelarou, Athina Patelarou

**Affiliations:** 1Department of Nursing, School of Health Sciences, Hellenic Mediterranean University, 71410 Heraklion, Greece; 2Department of Nursing, School of Health Rehabilitation Sciences, University of Patras, 26504 Patra, Greece; 3Department of Social Work, School of Health Sciences, Hellenic Mediterranean University, 71410 Heraklion, Greece

**Keywords:** COVID-19, nursing education, distance learning, e-learning, psychosocial adaptation, qualitative approach

## Abstract

(1) Background: The COVID-19 pandemic posed a major threat to global health and on the educational field. The purpose of this study is to identify and illustrate the psychosocial adaptation of nursing students to the sudden and exclusive application of distance learning during the COVID-19 pandemic; (2) Methods: A qualitative interview research has been designed. Two focus groups of seven members each and six individual interviews were conducted in a sample of undergraduate nursing students in Greece from 3 March 2021, to 9 April 2021.; (3) Results: A qualitative thematic analysis of the data identified six themes: 1. Challenges; 2. Concerns; 3. Social changes; 4. Negative Emotions; 5. Evaluation; and 6. Teaching strategies.; (4) Conclusions: During its implementation, it identified gaps and weaknesses in the entire academic community. The study of the psychosocial adaptation of the academic community is considered crucial, as it can highlight the individual difficulties in distance learning and contribute to the improvement of the methods of its the improvement of its methods.

## 1. Introduction

According to data from the World Health Organization (WHO), humanity faced the challenge of a new threat to public health: the COVID-19 pandemic. The high mortality rates of the COVID-19 pandemic, as reported by the World Health Organization (WHO), due to the rapid spread of the virus have forced governments around the world to take restrictive measures to manage this crisis [1]. However, one of the restrictive measures was social isolation, which forced the sudden transformation of the daily conditions of the general population on the whole. The elderly was initially forced to move away from their younger family members because of the risk they were facing by the COVID-19 pandemic [2]. Meanwhile, health professionals were faced with the harsh reality that called for them to leave their family for a long time due to the fear of spreading the virus [3]. Similarly, this abrupt change has led to the implementation of distance learning, revealing gaps and weaknesses in nursing students that are often not reflected in their true extent [4]. The aim of the present study is to illustrate the psychosocial adaptation of nursing students in the abrupt and exclusive application of distance learning during the COVID-19 pandemic.

Many difficulties were also found in the social adaptation of the academic population with the lack of interaction between its members, in the context of which mutual support and mental empowerment were cultivated, contributing positively to the socialization of students [5]. It is worth noting that distance learning has prospects for its participants [6]. Nevertheless, there is concern about the future professional development of members of higher education with the main focus of the sudden and exclusive use of distance learning during the COVID-19 pandemic [7]. The COVID-19 pandemic is a social phenomenon that brought social changes in the field of education with the exclusive use of distance learning. Nursing students felt uncertain about their future development and showed a deficit in willingness to participate in the educational process during the COVID-19 pandemic. As a result of resignation and isolation, while in several cases the combination of these two data can cause mental disorders in individuals [8].

An important element is the students’ concern about their future progression to the next levels. The inability to apply their clinical practice, which is a prerequisite for continuing their studies, was a decisive factor in increasing their anxiety. A recent study shows the belief of nursing students that missing out on their clinical practice will cause problems in their future career path [9]. The mental health of students was also affected by the lack of socialization and social interaction with their teachers. Distance learning with its sudden and exclusive use affected the interaction of educators with students resulting in the introversion and isolation of the former. It is worth noting that often these negative emotions also changed the learning outcome of the trainees [10]. On the other hand, the general view as it developed during the COVID-19 pandemic was decisively subversive for the daily life of people in modern societies, regardless of age. More specifically, economic effects were observed which changed student experience as a whole as many were forced to return to their paternal homes, while there was no longer the opportunity of face-to-face meetings with their peers [11]. At the same time, their only means of communication were social media platforms, while the physical presence of friends was replaced by virtual groups of distance learning [12].

## 2. Materials and Methods

This is a qualitative study of individual and focus groups interviews. The interview is the most appropriate method of data collection for a deeper approach to the topic under consideration. The attitudes and beliefs of the participants were examined, allowing the researcher to understand their views. The interviews were conducted via Skype because of the COVID-19 pandemic restrictive measures. However, data collection through individual and collective interviews enabled a holistic assessment of the issue [6]. The present study based on the phenomenon of saturation, as described by the literature, included 6 individual interviews and 2 focus groups of 7 members each. The interviews were conducted in focus groups which consisted of 7 members each. Participants were purposely sampled and recruited by the researcher (RA). The majority of participants were women (85.0%). Moreover, 45% were in the middle of their studies. In addition, 25% were students over 29 years old, as the majority of the participants were in the age group of 20–23.

Due to restrictive measures of the COVID-19 pandemic, the interviews were conducted remotely via SKYPE. Individual and group interviews were conducted from 11 March 2021 to 9 April 2021. Individual interviews were conducted by RA and lasted from 30 to 40 min, and were fulfilled until saturation was observed during data collection. Subsequently, focus group interviews were conducted and lasted from 2.15 to 2.30 min, as recommended by the literature [6]. The 2 focus interview groups comprised of 7 people per group, were attended by a qualified facilitator and a distance educator who were the researchers. The different interviewees were identified with different alphabetic codes. The abbreviations FG and IN were assigned to define focus group interviews and individuals’ interviews, respectively. Data extracted from focus groups 1 and 2 were indicated using (FG1) and (FG2).

The group chose the inductive method using a common reference of individual and collective interviews. Using an uncommon quotation can challenge the comparison of the results and cause possible disorientation. Furthermore, in the present study, indicative questions were used as a guide for the semi-structured interview and for the focus groups as well. Furthermore, interview guide questions were recommended according to the literature [13]. Last but not least, the questions started from a general context, followed by the customization of nursing students to the sudden and exclusive transformation of distance learning during the COVID-19 pandemic. The responses of the individual interview participants and the focus groups were colored to be classified into similar code categories. Comparisons were made between the data, and when new codes emerged, they were noted. The research team then searched for the most important codes, which were those most frequently mentioned, and thus formed a strong research plan.

### 2.1. Researcher’s Adaptability

At the beginning of the individual interviews, shyness was observed, while during the interview, the nursing students participating felt comfortable sharing their experiences. In addition, the respect and confidentiality shown by the team of researchers helped them to provide as much information as possible, to deepen the investigation of the research issue [14]. On the other hand, in focus group interviews, the power of group interaction was evident. As a result of this interaction, a wider range of information was provided. Moreover, the research team, in order to protect the credibility of the research, during the process of individual and group interviews, seriously took into account ethical issues trying to suppress their individual views and convictions.

### 2.2. Data Analysis

The direct involvement of the researchers in the data collection process greatly helped at the stage of analysis. Thus, after the recordings of the individuals’ and focus group interviews were completed, the first author discussed the interviews with RA. Both of them devoted themselves to data analysis, as was recommended by the literature, following the valid scientific analysis steps from collection to completion. Our initial research question, whether participants had had previous experience in distance learning, was used as an “ice breaking question” in order to reduce participants’ embarrassment, while the question of whether the participants responded easily to the sudden and exclusive use of distance learning during the COVID-19 pandemic was of great importance for the direction of the questions and the analysis of the data. Our analysis was based on specific questions of general context, while through the sub-questions that were asked, we approached the specific context, following the inductive method [15].

However, before starting the data analysis, the researchers deemed it necessary to read the updated literature on the research topic once again and as a next step they proceeded to the data analysis. At this stage, the research team carefully studied the notes regarding first impressions. Once this step was completed, the coding process began. In each section of data, subtitles, keywords, phrases, and sentences were defined. The categorization of the findings was based on the relevance and research interest that emerged from the data (e.g., technological resources, distance learning evaluation, future course, clinical practice, and mental and social dimensions of the phenomenon).

It was observed that the data of volunteering, which were provided, were inadequate for this subject. Initially, there was the expectation that clinical practice and practical experience would be categorized under two different codes. Nevertheless, they created a dominant joint code, the one of clinical practice. More specifically, the thematic map includes the technological resources, the response to the transition from face-to-face learning to the distance learning assessment of nursing students, etc. The purpose of the thematic map was the brief presentation of the main and secondary topics. The research team recorded a detailed report of the data, having already agreed on the final axes of the research topics. In this way, it enhanced the reliability and validity of the main findings highlighted by the present study [5].

In order to ensure the validity and reliability of the research, a third researcher checked the correlation of the written analysis with the interview data. Similarly, he reviewed the variability of the process from the collection of data to the conduct of the findings of the present research. The main criterion for the inclusion of students was that those with at least one academic year of study, while those who did not participate in the educational process during the period of implementation of the restrictive measures of the COVID-19 pandemic were excluded. The findings from the first part of the study were based on the attitude and views of nursing students about distance learning. They were also used to stimulate discussion and encourage participants to express their concerns. The nursing students’ concerns will reflect in their psychosocial adaptation to the new type of learning during the COVID-19 pandemic (Table 1), but do not interpret the results or discuss their implications.

Additionally, it is worth noting that the Ethical committee of the Hellenic Mediterranean University (No 37/15/12/20) permitted and approved the present study. The participants were asked to complete a consent form before the interview. After receiving the participants’ written consent, the researchers shared with them an information leaflet with details about the way in which those individual and group interviews would be conducted.

## 3. Results

A total of 6 individuals and 2 focus groups comprised of 7 participants were included in the study, The majority of participants were women (85.0%), while 45% were in the middle of their studies. In addition, 25% were students over 29 years old, as the majority of the participants were in the age group of 20–23.

### 3.1. Challenges and Measures to Adapt to Sudden, Unexpected and Rapid Transition to Online Learning

Nursing students’ reports of their response to the sudden shift to distance learning during the pandemic period revealed three subcodes. The first subcode was the previous experience. Interviewees with no previous experience in distance learning found it difficult to adjust to it during the sudden transition of the COVID-19 pandemic. IN: *‘… I had no experience in distance learning before. I didn’t even know what the meaning of this learning type was*.’ (IN). The data analysis revealed the family changes as the second subcode. Several changes have taken place in families because of the pandemic. Many people were unemployed, and this new condition brought difficulties to the family balance. FG: *‘My father is a taxi driver … income went to zero during the quarantine because of COVID-19 pandemic.’* (FG2). Likewise, several students returned to their homeland, and this was a challenge because they had to keep up with the needs of the other members. FG: *‘I returned to my paternal home.’* (FG1). However, many students stated that distance learning offers them not only more time for deeper thoughts, but also the opportunity to be with their families. FG: *‘My father suffers from a very serious illness. Returning to our parental home gave us the opportunity to take care of him. So … we’ve reduced our costs.’* (FG1). At last, the third subcode showed familiarity with New Technology. There was less familiarity in the use of new technology for older teachers than younger teachers. Despite that, the majority of participants agreed, it was a challenge for everyone. FG: *‘The older the teacher, were more unfamiliar with the use of technology because they felt the lecture became more difficult. While the youngest teachers were familiar with new technology, they were already trained in it.’* (FG1).

### 3.2. Concerns for the Future Progress

The second subcode was about the adequacy of resources. The equipment or technological resources are considered necessary for the conduct of distance learning. However, it was not taken for granted that all students had it during the sudden transformation of in person to distance learning. FG: ‘*I did not have a computer; I got a small tablet that did not cater for my needs. I had many problems*.’ (FG1). Academic institution support: However, the university seems to have attempted to cover the needs of nursing students with no financial burden, and this is acknowledged by the majority of interviewees. FG: ‘*In the beginning, there were some issues, however, we had technical support by the IT of the institution*’. Unfortunately, the difficulties in connecting to the internet were many. To begin with, there was a participation limit for students per online lecture. In addition, disconnection issues were apparent. IN: ‘*I had a lot of problems in the beginning, especially when the platform did not respond because of the connection with the server; as a result, there were many moments when I disconnected from the lesson*.’ (IN).

### 3.3. The Social Dimension

Students’ interaction with teachers and their classmates was eliminated due to the sudden and exclusive use of distance learning during the pandemic. FG: ‘…*But the interaction with teachers and fellow students is completely lost… This is something you were missing*.’ (FG2). There is a gap in communication between teachers and students. IN: ‘*I’m ashamed to turn on the microphone and say what I did not understand so there is a gap*.’ (IN). While distance learning offered students the opportunity to communicate more: IN: ‘*During distance learning to-face learning, the lesson becomes much more interactive*.’ (IN). Use of social media. The interviewees stated that the only way to communicate with their fellow students and teachers was through the use of social media during the COVID-19 pandemic.IN: ‘*We no longer see our fellow students. Relationships are now impersonal, because we live in different cities while previously, we had a common city. Now, we can only get through to each other using our mobile through social media. It is not the same. There’s no other way to communicate*.’ (IN). Opportunities for socialization: An important statement of the students was that during the pandemic, there were no opportunities for their socialization. FG: ‘*The interaction was reduced, and the socialization for me was nonexistent* (FG1).

### 3.4. The Mental Dimension

Negative emotions: The interviewees appeared to be mentally vulnerable owing to the quarantine imposed because of the COVID-19 pandemic. FG: ‘The emotions that dominated both the quarantines and the distance learning were negative’. (FG2). FG: ‘…*I felt unproductive, in the first quarantine. It was very difficult for me. I asked the support of a psychologist and a psychiatrist; I could not accept to the new conditions…*’ (FG1). However, nursing students felt uncertain about the duration of quarantine, while appearing to have no other mental reserves. FG: ‘…*The uncertainty about how long it will last is hard to bear. I cannot stand it anymore* …’ (FG2). Ιn addition, creativity was absent from the lives of nursing students resulting in mental disorders and their gradual resignation. FG: ‘…*For me, it is permanent fatigue. I find no reason to get out of bed. The only thing that will make me get up is that I have a lesson. I have no motivation…. Sleep disorders*.’ (FG2).

Fear of volunteering: The volunteering for many participants was linked to the absence of clinical practice due to the COVID-19 pandemic. The high rate of coronavirus spread was the reason why many students were afraid to risk their lives through volunteering. IN: *‘Maybe … I had thought of volunteering at the beginning of the 1st lockdown … but the clinics were filled with patients, so it was also a greater risk … for my health…*’ (IN). Abandonment of studies. The restrictive measures, especially the social distancing imposed because of the COVID-19 pandemic, burdened the mental health of nursing students. In addition, moving away from their daily activities pushed them to the extremes. As a result, many of them wanted to move away or abandon their studies. FG: “… *For me, it is permanent fatigue to participate in my studies by distance.. The only thing that will make me get up is that I have class and for that I have no motivation*….” (FG2).

An important point of the study was the concern of nursing students about their future courses. The majority of participants felt insecure about their future, as important pillars of the science of nursing, such as evaluation, clinical practice, and performance, were at stake.

### 3.5. Students’ Evaluation

The assessment of distance learning has proven to be stressful because of ineffective internet connectivity. FG: ‘…*The distance learning exams were upsetting because there was always this fear that the system would fail*.’ (FG1). Similarly, the time of examination was insufficient for the students in comparison to distance learning assessment. FG: *‘… As for the exam, I was worried about the time we were allocated’*. (FG2). Although students were required to participate in distance learning assessment, they do not seem to accept its effectiveness as they did in face-to-face learning conditions. IN: ‘*I believe that it does not have the reliability that distance learning evaluation has’*. (IN). Changes in students’ performance. There was a decline in students’ performance during distance learning. IN: ‘*Generally my performance was relatively at the same level, although it decreased slightly throughout distance learning*’. (IN). Similarly, several interviewees had a drop in their performance due to the fact that they felt ashamed to participate in the distance learning process. FG: *Many times I thought I would like to ask questions in the online lecture. But I was embarrassed…and chose not to participate*.’ (FG2).

Clinical practice: Undoubtedly, the science of nursing is linked to its practical application. Respondents were particularly concerned about not achieving a combination of theoretical and practical context through distance learning during the pandemic. FG: ‘*In Nursing Science, the clinical presentation is indispensable. Face-to-face learning is necessary, laboratories are necessary. It’s not possible to be a nurse without them*.’ (FG1). The majority of the participants were convinced that their removal from clinical practice would cause problems in their future careers. IN: ‘*Actually, during face-to-face learning we had the opportunity to visit hospitals and do the clinical practice. Now, by not doing our clinical practice, I feel many shortcomings in my knowledge’*. (IN).

Challenges and prospects of distance learning: Respondents identified opportunities and challenges in distance learning. Some of the participating nursing students were employed in parallel with their studies. They felt that they had the opportunity to do that due to the time flexibility owing to distance learning. FG: *‘It was tiring for me to do the same route for so many years in order to be in time for the lecture… Through distance learning, I saved money and effort*.’ (FG1). At the same time, there was the opinion of some participants who opposed the implementation of distance learning as they considered that it limited and removed them from their educational life. FG: ‘*This entire process of distance learning has left us behind with our plans and does not let us progress as we would like to. Undoubtedly, I think that personal contact is much preferable than living at a distance learning*’ (FG2). It is stated that the difference of opinion of the participants in relation to distance learning depends on their individual obligations. FG: ‘*As an employee, working from a distance learning has some advantages, but as a student, I think it affects the evolution of humans.*’ (FG2).

### 3.6. Effective Teaching Strategies

The majority of participants identified interaction between teachers and fellow students as the greatest asset and also a higher quality of learning in of learning in face-to-face education. The majority of participants identified interaction between teachers and peers as the greatest advantage in face-to-face learning. IN: ‘‘*Face-to-face learning is the best type of teaching and I think I would not replace it for any other form of teaching. I think it is the most important and interactive thing that can exist in education*.’ (IN). Combination of both teaching methods: Nursing students seem to approve of the blending of the two teaching methods while arguing that the practical framework should be conducted exclusively through face-to-face learning. IN: *‘I would like the practical part to be done through face-to-face learning, while in theory I think that distance learning might also help to increase the participation of my fellow students who are working alongside their studies*.’ (IN).

## 4. Discussion

The difficulty of adapting students without prior experience with distance learning has emerged in the current study. In addition, new research findings have revealed that the smooth execution of educational objectives must be achieved due to the lack of previous exposure of the participants in its exclusive use [16]. However, this sudden change in exclusive use of distance learning was so intense that the difficulty focused mainly on the degree of difficulty of accessing any type of platform, as noted in a new research report [17].

Dependent factors, such as unemployment, income, and quality production among members of each family, have emerged. Following this position, a change in family income has emerged due to COVID-19 pandemic, which affects the willingness of nursing students to participate in electronic learning [18], while 25.7% of the participants have experienced intense stress and fear because of the family changes caused by the COVID-19 pandemic, which affects their learning performance [19]. The majority of participants in the present study agreed that adapting to distance learning and becoming familiar with new technologies was a challenge for all. While the majority of respondents agreed that familiarity with new technologies is commensurate with the age of students and teachers. The difficulty of adapting to distance learning during the COVID-19 pandemic because of users’ unfamiliarity with new technology and online tools was reported by the findings of recent research [20]. In contrast, most of the participants (95%) in a medical research study in China showed their satisfaction with implementing distance learning because of their familiarity with new technology and their previous experience with distance learning [21].

Without a doubt, technological resources are considered necessary for the delivery of distance learning. In the present study, it was noted that several of the respondents did not have the necessary technology resources and they encountered difficulties connecting to the internet. On the other hand, they stated that they had the support of the academic institution, which tried to meet the needs of the nursing students.

As a result, recent studies have revealed the financial burden placed on families in order for their children to participate in distance learning [22]. Meanwhile, one more study stressed that internet connectivity had serious problems, especially in remote rural areas [23]. However, the best way to deal with challenges and technical difficulties was highlighted by a study in academic institutions in developed countries where despite the efforts of teachers, students, and academic institutions, it was stated that none of them was prepared for the unanticipated and exclusive application of distance learning during the COVID-19 pandemic [24]. The present study showed a gap in students’ communication with their tutors because of the sudden and exclusive use of distance learning during the COVID-19 pandemic. Similarly, recent studies have identified the same gap, highlighting deep concerns about the level of interaction concerning face-to-face learning [25,26].

Additionally, it is worth noting that the findings of our study about the lack of social interaction and the insecurity that this situation may cause to the members of the academic community are associated with the Greek culture. In detail, in the present study Greek nursing students felt the social isolation caused by the COVID-19 pandemic more strongly, as they could not keep tight bonds with their co-students and teachers in the academic environment. As revealed by the ethnography and anthropology, individuals in the Greek society keep close ties with members of the same social unit and they have high needs of inclusion and approval by the group. Researchers state that in Greece, the concept of the in-group is central and it is defined as a community which provides protection, social security, and a warm environment [27]. In a period of crisis, it was hard to keep those bonds in the academic community tight; nevertheless, Greeks have always placed a high value on education and embraced learning. Additionally, as mentioned in the literature of the classical Greeks of antiquity, such as Aristotle, a basic condition of the sustainability of democracy and civilization is the deep connection of learning with intellectual innovations [28].

According to the results of the present study, the majority of participants supported the use of social media as the only means of communication and socialization. Likewise, the positive role of social media during the COVID-19 pandemic was recently recognized [29]. In addition, a specific focus was given in another study, social media has often emerged as a provider of large amounts of information with dubious content, leading to panic in times of crisis, such as the COVID-19 pandemic [30]. Another study noted the WHO recommendation regarding a new phenomenon called “Infodemic” and needs appropriate attention [31]. Similarly, in the present study, most nursing students stated that during the COVID-19 pandemic, there were no opportunities for socialization. It is worth noting that, during the COVID-19 pandemic, more opportunities for parental socialization with their children were observed in order to support their negative emotions, as presented in the literature in a recent research study in Italy [32]. However, a study conducted in China identified a lack of socialization opportunities in social distancing as a determining factor in creating negative emotions for students during the COVID-19 pandemic [33]. It is worth noted that the present study, the majority of respondents reported being mentally vulnerable due to quarantine because of the COVID-19 pandemic, while many of the participants felt they did not have mental reserves as creativity was absent from their lives during this period.

At the same time, in the present study, the issue of volunteering was raised, with the majority of participants expressing fear of the spread of the virus in the case of their participation. In contrast, in a study in Saudi Arabia, health students said they were willing to volunteer to treat the virus COVID-19 [34]. However, the majority of participants of 61.4% in a study in Vietnam showed their interest in volunteering, but not in the field of healthcare [35]. In the present study, the majority of participants experienced mental disorders due to quarantine and recurrent lockdowns that often resulted in deviations from their learning goals. It is worth noting that a recent study of students at the University of Washington School of Dentistry highlighted that 38.23% reported mental health problems and expressed the intention to drop out of school [36]. Furthermore, the findings of another research in Poland of students of various scientific backgrounds found that 76.96% of respondents developed psychopathological symptoms during the COVID-19 pandemic [37], while as stated in a study conducted in Albania, the mental health of students (50%) and their family members (51.6%), was imperiled during the period of quarantine [38]. Furthermore, a recent Israeli study found that the COVID-19 pandemic caused stress and other negative emotions in students, with many resorting to substance use [39]. During the COVID-19 pandemic, the sudden and exclusive use of face-to-face teaching in the exclusive use of distance learning had a significant impact on nursing students’ future courses.

In this study, participants said they were insecure about their future as serious fields, such as clinical practice, evaluation, and performance, were affected. According to recent research, students’ mental health has been negatively affected concerning their future course, with the majority of respondents (83%) appearing stressed during the COVID-19 pandemic [40]. However, in China, the findings of a recent research appeared that a telephone support line was set up for students to provide them with the appropriate mental empowerment in the face of a sudden change in education. In the same study, the university institution presented an effort to meet students’ needs interactively, highlighting the shortcomings of distance learning as opportunities to accelerate development [41]. Nonetheless, the current study highlighted an important issue that concerned the majority of participants regarding clinical practice implementation. Nursing students during social distancing were forced to postpone clinical practice, resulting in intense concern about their future adequacy. Although, students complained about the successful completion of their studies, while a smaller percentage of participants said they were willing to participate in clinical practice even at the risk of contracting the virus COVID-19, as it was focused by recent research in Canada [42]. Furthermore, an intercultural study found that nursing students had higher levels of stress for clinical practice during the COVID-19 pandemic than they had before [43]. It is worth mentioning that in the present study, the majority of participants identified challenges in distance learning as well. Many of the participants mentioned their satisfaction with the flexibility offered by distance learning. Similarly, the advantage of flexibility in distance learning as it was afforded by respondents (32% of the total) has been recognized [44]. However, in the present study, the majority of participants appeared willing to accept a blended learning system of education. Despite that, they wished the practical application of their studies would be face-to-face learning. Notably, current research on pediatric students showed that many respondents regarded blended learning as effective because it combined practical application through face-to-face learning and distance learning, which could provide the opportunity to learn innovative educational platforms [45].

In addition, in the present study, the majority of nursing students stated that during COVID-19 pandemic, there were no opportunities for socialization. In addition, according to a recent bibliographic study in which the experiences of nursing students during the COVID-19 pandemic was presented, there was a strong connection between the negative effects on the mental health of nursing students and the feeling of stress. Thus, their expectations were negatively altered in relation to their sustainability and future development [44]. The findings highlighted the particular difficulties that occurred in conducting the educational process because of the abrupt transition from face-to-face to distance learning, especially for nursing students. However, these difficulties concerned the wider educational sector as well, as it is a conduit for all the applications. Moreover, they contributed to the collection of valuable information through the experiences of the participants. In this subject, it is worth noting that the information in this study is recommended for further scientific investigation to improve and develop the methods of using distance learning.

### Strengths and Limitations

To the best of our knowledge, this is the first study investigating this important topic in a sample of Greek nursing students. Moreover, the qualitative study design and the combination of both individual interviews and focus groups were adequate techniques for investigating in depth the psychosocial impact of the COVID-19 pandemic on the nursing students’ sample. Additionally, this research design allows the researchers to explore the participants’ views and experiences not only one-to-one, but in a pattern of interaction as well. On the other hand, a limitation of the present study was the number of institutions participating, as a multicenter study would provide stronger evidence for the academic community and would offer to the researchers the possibility to make further comparisons among the sample. Overall, further scientific research is needed to explore potential solutions for the smooth psychosocial adaptation of nursing students to the new digital education context.

This study is not without limitations. A methodological limitation was the sample size, which proved to be satisfactory, as saturation was observed in the sample responses. Due to the saturation of individual interviews, focus group interviews were followed for maximum reliability of the study. Another limitation was the number of institutions participating, as a multicenter study would provide stronger evidence for the academic community and would offer to the researchers the possibility to make further comparisons among the sample. Overall, further scientific research is needed to explore potential solutions for the smooth psychosocial adaptation of nursing students to the new digital education context.

## 5. Conclusions

In conclusion, the emergence of the virus COVID-19 has caused a crisis in the global community. In particular, it affected the field of health and education, among others. Governments all over the world acted appropriately by taking restrictive measures to reduce the rapid spread of the virus. Thus, in the sector of education, there was a sudden and exclusive use of distance learning, replacing face-to-face learning. Based on the findings of this study, further research is recommended in order to improve and develop the most appropriate methods of using distance learning to ensure the development of a new sustainable educational reality.

## Figures and Tables

**Table 1 healthcare-11-01178-t001:** Thematic map—sections and subsections.

Challenges and measures to adapt to sudden, unexpected and rapid transition to online learning	Previous experience
	Family changes
	Familiarity with New Tech
Concerns for the future progress	Adequacy of resources
	Academic institution
	Technical difficulties
The social dimension of the phenomenon with reference to the interaction among teachers and students	Interaction with teachers and classmates
	Use of Social Media
	Opportunities for socialization
The mental dimension of the phenomenon with reference to the negative emotions of the students	Negative emotions
	Fear of volunteering
	Abandonment of studies
Students’ evaluation	Evaluation
	Performance change
	Clinical practice
Effective teaching strategies	Challenges and prospects of distance learning
	Benefits for Face-to-Face learning
	Combination of both teaching methods

## Data Availability

The datasets used and/or analyzed during the current study are available from the corresponding author on reasonable request.

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
