# Peer review of "Distance Learning in Nursing Education during the COVID-19 Pandemic: Psychosocial Impact for the Greek Nursing Students—A Qualitative Approach"

_healthcare, 2023, doi:10.3390/healthcare11081178_

Round 1
Reviewer 1 Report
The submission presents a valid contribution to the field for which the co-authors have offered insights. On the other hand, the co-authors ought to consider the cultural context of the Greek nursing students surveyed for this research. What specific aspects of Greek culture could prove relevant in the findings of the paper? In other words, we hope that the co-authors consider an interdisciplinary discussion that would incorporate fields like (for instance) ethnography, anthropology, and cultural studies. How does the ethnic/cultural Greek background of the surveyed students influence the ways in which the students experience distance learning? Two possible ways of answering this question would require some interdisciplinary readings. In the first possible way, the co-authors can make a case for how the Greek culture of the students has little influence in the student feelings about distance learning; the internationalized and globalized atmosphere of nursing education precludes the impact of national or ethnic cultures. In the second possible way, the co-authors can argue for some influence of Greek culture.
Author Response
Response to Reviewer 1 Comments
Point by point
We would like to warmly thank the reviewers for their kind words and valuable suggestions which have significantly contributed to the improvement of this manuscript. All our revisions have been marked in red in the revised manuscript.
Dear authors,
Thank you for the opportunity to review your interesting and well written manuscript. I will give my feedback following the structure of the manuscript.
Point 1. General
The submission presents a valid contribution to the field for which the co-authors have offered insights. On the other hand, the co-authors ought to consider the cultural context of the Greek nursing students surveyed for this research. What specific aspects of Greek culture could prove relevant in the findings of the paper? In other words, we hope that the co-authors consider an interdisciplinary discussion that would incorporate fields like (for instance) ethnography, anthropology, and cultural studies. How does the ethnic/cultural Greek background of the surveyed students influence the ways in which the students experience distance learning? Two possible ways of answering this question would require some interdisciplinary readings. In the first possible way, the co-authors can make a case for how the Greek culture of the students has little influence in the student feelings about distance learning; the internationalized and globalized atmosphere of nursing education precludes the impact of national or ethnic cultures. In the second possible way, the co-authors can argue for some influence of Greek culture.
Response 1: Thank you very much for the kind suggestion. The suggested comment and the references are discussed in the revised version of the manuscript as requested (lines 434-450).

Reviewer 2 Report
I commend the authors for undertaking the current study. However, I have several concerns which I would recommend the authors to address.
General comments: Many sentences in the manuscript are not clear. The English language is poor, especially in the ‘discussion’ section. In some instances, “COVID-19 pandemic” is used, while in other instances “pandemic COVID-19” is used. Pls standardize the use of this phrase. Pls change ‘distance education learning’ to ‘distance learning’ or ‘distance education’ throughout the manuscript.
Introduction
1. Line 29 – Pls use the abbreviation for ‘World Health Organization’ with ‘WHO in brackets or use the full form of ‘WHO’ in line 357.
2. Lines 30-33 – You mention the mortality rate stated by Johns Hopkins University and then go on to cite using a totally unrelated reference (reference 2, which is about development and validation of a questionnaire). Pls change the self-citation and cite from valid sources like the World Health Organization, Johns Hopkins University, etc.
3. Line 42 – Pls use either ‘distance education’ or ‘distance learning’, not ‘distance education learning’.
4. Lines 53-55 – It is not clear what you want to say.
5. Lines 60-61 – Do you mean ‘socialization and social interaction with their teachers’ or ‘lack of socialization and social interaction with their teachers’?
6. Lines 64-71 – Pls rewrite. The English language is not clear. Moreover, in line 70, you mention ‘physical presence of friends. Is it ‘presence’ or ‘absence’?
Materials and methods
Lines 85-86 – This sentence has already been mentioned above in lines 76-77.
Results
Why are the subsections under the ‘results’ section with the same title, and what is the point of dividing into several subsections when they all have the same title?
1. Lines 159-160 – Pls explain what you mean by this sentence or rewrite it.
2. Lines 160-163 – This should be in the ‘materials and methods’ section.
3. Line 175 – You say that “distance education learning during the pandemic period revealed three subcodes”. Pls make it clear what the 3 subcodes are.
4. Lines 235-236 – What do you mean by this sentence?
5. Lines 239-241 and 249-241 are the same.
6. Lines 260-263 - What do you mean by this sentence?
7. Lines 264 – 265 - What do you mean by this sentence?
8. Lines 298-300 - What do you mean by this sentence?
Discussion
The discussion is not focused and requires major revision.
1. Lines 313-315 - What do you mean by this sentence?
2. Line 316 – What do you mean by FAM-Ely?
3. Lines 369-387 – This whole paragraph needs rewriting. Moreover, what is the relationship between volunteering and distance education? Why was this included as a factor? Pls explain.
4. Lines 408-409 – “a high rate of respondents 408 (32% of the total)”. How do you consider 32% as high rate?
5. Lines 426-430 – These lines are not appropriate in the conclusions. The conclusions should highlight the main findings of the study and may be provide important recommendations.
Author contributions
Line 440 – ‘G.M. and’ who?
References
Most references need to be rechecked. Some of the journal names are in abbreviations and some not. Some journal names have only the first alphabets of each word. The format of writing the year, volume, etc. also need to be rechecked thoroughly.
Author Response
Response to Reviewer 2 Comments
Point by point
We would like to warmly thank the reviewers for their kind words and valuable suggestions which improve significantly this paper. We have followed them strictly in order to improve our manuscript. All our revisions are with track changes and marked in yellow in the revised manuscript.
Dear Author’s
Point 1: General comments
Many sentences in the manuscript are not clear. The English language is poor, especially in the ‘discussion’ section. In some instances, “COVID-19 pandemic” is used, while in other instances “pandemic COVID-19” is used. Pls standardize the use of this phrase. Pls change ‘distance education learning’ to ‘distance learning’ or ‘distance education’ throughout the manuscript.
Response 1: Many thanks for your comments. In the whole manuscript we have standardize the use of phrase COVID-19 pandemic, as you recommend. In addition, an English professional editing has been made on the manuscript.
Point 2: Introduction
- Line 29 – Pls use the abbreviation for ‘World Health Organization’ with ‘WHO in brackets or use the full form of ‘WHO’ in line 357.
- Lines 30-33 – You mention the mortality rate stated by Johns Hopkins University and then go on to cite using a totally unrelated reference (reference 2, which is about development and validation of a questionnaire). Pls change the self-citation and cite from valid sources like the World Health Organization, Johns Hopkins University, etc.
- Line 42 – Pls use either ‘distance education’ or ‘distance learning’, not ‘distance education learning’.
- Lines 53-55 – It is not clear what you want to say.
- Lines 60-61 – Do you mean ‘socialization and social interaction with their teachers’ or ‘lack ofsocialization and social interaction with their teachers’?
- Lines 64-71 – Pls rewrite. The English language is not clear. Moreover, in line 70, you mention ‘physical presence of friends. Is it ‘presence’ or ‘absence’?
Response 2: We thank the reviewer for this valuable comments.
- Lines 32-33: We have used the abbreviation for ‘World Health Organization’ as you noted.
- Lines 32-33: We have corrected the cite using valid sources like the World Health Organization.
- Line 41: We have corrected “distance learning” as you suggested.
- Lines 53-53: The phrase has been revised in order to be clear.
- Lines 65-66 – We mean ‘lack of socialization and social interaction with their teachers’.
- Lines 65-72 – We have revised the text in order to define better the meaning in the English language. Moreover, in lines 77-78, we mention that the ‘physical presence of friends was replaced by virtual groups of distance learning”.
Point 3: Materials and methods
Lines 85-86 – This sentence has already been mentioned above in lines 76-77.
Response 3: We feel thankful for your comment. It has been corrected as you suggested (Lines 85-86 ).
Point 4: Results
Why are the subsections under the ‘results’ section with the same title, and what is the point of dividing into several subsections when they all have the same title?
- Lines 159-160 – Pls explain what you mean by this sentence or rewrite it.
- Lines 160-163 – This should be in the ‘materials and methods’ section.
- Line 175 – You say that “distance education learning during the pandemic period revealed three subcodes”. Pls make it clear what the 3 subcodes are.
- Lines 235-236 – What do you mean by this sentence?
- Lines 239-241 and 249-241 are the same.
- Lines 260-263 - What do you mean by this sentence?
- Lines 264 – 265 - What do you mean by this sentence?
- Lines 298-300 - What do you mean by this sentence?
Response 4: We thank the reviewer for this valuable comments. We have made the corrections at the subsections point by point.
- Lines 157-159 – We have rewritten the sentence as you noted in order to have a clear meaning.
- Lines 160-163 – It has been changed in the section of ‘materials and methods’ section, (lines 125-127).
- Lines 233, 243,253 – We have noted three subcodes with clarity, as you suggested.
- Lines 235-236 – The suggested comment is defined in the revised version of the manuscript as requested (lines 242-243).
- Lines 239-241 and 249-241 are the same.: We have made the correction as you suggested.
- Lines 260-263 - The suggested comment is defined in the revised version of the manuscript as requested (lines 253-254).
- Lines 264 – 265 - The suggested comment is defined in the revised version of the manuscript as requested (lines 256-258).
- Lines 298-300 - The suggested comment is defined in the revised version of the manuscript as requested (lines 302-303).
Point 5: Discussion
The discussion is not focused and requires major revision.
- Lines 313-315 - What do you mean by this sentence?
- Line 316 – What do you mean by FAM-Ely?
- Lines 369-387 – This whole paragraph needs rewriting. Moreover, what is the relationship between volunteering and distance education? Why was this included as a factor? Pls explain.
- Lines 408-409 – “a high rate of respondents 408 (32% of the total)”. How do you consider 32% as high rate?
- Lines 426-430 – These lines are not appropriate in the conclusions. The conclusions should highlight the main findings of the study and may be provide important recommendations.
Response 5: We thank the reviewer for this valuable comments.
- 1. Lines 313-315 - The suggested comment is defined in the revised version of the manuscript as requested (lines 402-405).
- Line 316 – What do you mean by FAM-Ely? It has been removed (line 406).
- Lines 369-387 – The suggested comment is defined in the revised version of the manuscript as requested (lines 479-497).
- Lines 408-409 – “a high rate of respondents 408 (32% of the total)”. How do you consider 32% as high rate? The suggested comment is defined in the revised version of the manuscript as requested (line 522).
- Lines 426-430 – These lines are not appropriate in the conclusions. The conclusions should highlight the main findings of the study and may be provide important recommendations. : The suggested comment is defined in the revised version of the manuscript as requested (lines 536-543).
Point 6: Author contributions
Line 440 – ‘G.M. and’ who?
Response 6: Many thanks for your comment. It has been revised in the text in Line 564 as ‘G.M. and E.K.’
Point 7: References
Most references need to be rechecked. Some of the journal names are in abbreviations and some not. Some journal names have only the first alphabets of each word. The format of writing the year, volume, etc. also need to be rechecked thoroughly.
Response 7: Many thanks for your comment. The bibliographic list has been checked in detail in order to correct all the mistakes as you suggested.

Reviewer 3 Report
In this article, the authors study the psychosocial impact of distance learning in e Greek nursing education during the Covid-19 pandemic.
The title of the article is apposite but maybe to long (is the "qualitative approach" really needed in the title?); the keywords are well chosen; the abstract is clear and appropriate in length and content.
The introduction is sufficient to provide the necessary background. The references are relevant and recent.
Materials and methods' description is clear and detailed (even too much: things as this "However, before starting the data analysis, we deemed it necessary to read the updated literature on the research topic once again. Then we started analyzing the data. We read our data over and over again before moving on to the coding stage..." may be summarized).
The table layout and labelling is clearly readable.
The discussion of data is lucid and the conclusions are consistent.
The number of the interviewed subjects is limited, the novelty of the study is not so high, but still the scientific soundness of this research is solid and the potential interest for the readers of the journal is probably enhanced by the topicality and relevance of studies about the consequences of Covid-19 pandemic. So in my opinion it is, all considered, well suited for publication (after the suggested editing).
Author Response
Response to Reviewer 3 Comments
We would like to warmly thank the reviewers for their kind words and valuable suggestions which improve significantly this paper. We have followed them strictly in order to improve our manuscript. All our revisions are with track changes and marked in yellow in the revised manuscript
Dear authors,
Thank you for the opportunity to review your interesting and well written manuscript. I will give my feedback following the structure of the manuscript.
Point 1. General
In this article, the authors study the psychosocial impact of distance learning in e Greek nursing education during the Covid-19 pandemic.
Response 1: We thank the reviewer for this valuable comment.
Point 2. Title
The title of the article is apposite but maybe to long (is the "qualitative approach" really needed in the title?); the keywords are well chosen; the abstract is clear and appropriate in length and content.
Response 2: We thank the reviewer for the kind comment. In the section of the Instructions for authors, it was mentioned that the title of the manuscript should identify if the study reports (human or animal) trial data, or is a systematic review, meta-analysis or replication study.
Point 3. Introduction
The introduction is sufficient to provide the necessary background. The references are relevant and recent.
Response 3: Many thanks for your precious comment.
Point 4. Materials and methods
Materials and methods description is clear and detailed (even too much: things as this "However, before starting the data analysis, we deemed it necessary to read the updated literature on the research topic once again. Then we started analyzing the data. We read our data over and over again before moving on to the coding stage..." may be summarized). The table layout and labelling is clearly readable.
Response 4: We warmly thank the reviewer for this valuable comment which gave us the opportunity to summarized the information of data. (Line
Point 5. Discussion
The discussion of data is lucid and the conclusions are consistent.
Response 5: We warmly thank the reviewer for this valuable comment.
Point 6. Overall
The number of the interviewed subjects is limited, the novelty of the study is not so high, but still the scientific soundness of this research is solid and the potential interest for the readers of the journal is probably enhanced by the topicality and relevance of studies about the consequences of Covid-19 pandemic. So in my opinion it is, all considered, well suited for publication (after the suggested editing).
Response 6: We thank the reviewer for the kind comment

Reviewer 4 Report
The subject of the article is very interesting, and the study objective is important. The findings are eminent to the nursing profession and its research.
Some issues need to be addressed:
1. Table 1 is redundant. Perhaps it is better just to specify the characteristics such as school year and whether the student is working or not, and present age using average and range.
2. The first paragraph in the results chapter should be in the method chapter.
3. It is advisable to reorganize the results section. I suggest that each subchapter be given a title according to what is indicated in Table 2. The titles should be highlighted, and the quotes written in a different font to make it clearer to the reader.
For example:
3.2 Concerns for the future progress
Adequacy of resources… FG: ‘I did not have a computer; I got a small tablet that did not cater 199 for my needs. I had many problems.’(FG1).
4. Limitations of the study are not addressed.
5. There is no adequacy between some of the items indicated in the body of the article and their number in the bibliographic list. As for example: Item number 36 in line 384 appears in the list as item number 34. Item number 38 does not appear in the article at all.
6. There is a need of English professional editing (for example: row 139-140; rows 321-323).
Author Response
Response to Reviewer 4 Comments
We would like to warmly thank the reviewers for their kind words and valuable suggestions which improve significantly this paper. We have followed them strictly in order to improve our manuscript. All our revisions are with track changes and marked in yellow in the revised manuscript
Point by point
Dear authors,
Thank you for the opportunity to review your interesting and well written manuscript. I will give my feedback following the structure of the manuscript.
Point 1. General
The subject of the article is very interesting, and the study objective is important. The findings are eminent to the nursing profession and its research.
Response 1: Thank you very much for your comments.
Point 2. Table
Table 1 is redundant. Perhaps it is better just to specify the characteristics such as school year and whether the student is working or not, and present age using average and range.
Response 2: Thank you very much for your comments. Kindly note that the Table 1 was deducted. We have specified the characteristics such as school year and whether the student is working or not, and present age using average and range. The suggested comments are discussed in the revised version of the manuscript as requested (lines 215-219).
Point 3. Results
The first paragraph in the results chapter should be in the method chapter.
It is advisable to reorganize the results section. I suggest that each subchapter be given a title according to what is indicated in Table 2. The titles should be highlighted, and the quotes written in a different font to make it clearer to the reader.
For example:
Concerns for the future progress
Adequacy of resources… FG: ‘I did not have a computer; I got a small tablet that did not cater 199 for my needs. I had many problems.’(FG1).
Response 3: Many thanks for your comment. We have made the change of the first paragraph in the results chapter in the method chapter (lines 173-180). We have added subtitles on ach subchapter according to what is indicated in Table 2. and the quotes were written in a different font to make it clearer to the reader.
Point 4. Limitations
Limitations of the study are not addressed.
Response 4: Many thanks for your comment. Limitations of the study were addressed (lines 538-558).
Point 5. References
There is no adequacy between some of the items indicated in the body of the article and their number in the bibliographic list. As for example: Item number 36 in line 384 appears in the list as item number 34. Item number 38 does not appear in the article at all.
Response 5: We thank the reviewer for this valuable comment which has given the opportunity to checked in detail the bibliographic list in order to correct the adequacy between all of the items indicated in the body of the article.
Point 6. English professional editing
There is a need of English professional editing (for example: row 139-140; rows 321-323).
Response 6: Many thanks for your comment. An undergo extensive English revisions have been made on the manuscript

Round 2
Reviewer 2 Report
The authors have addressed my concerns.